# Efficient CDF Approximations for Normalizing Flows

**Chandramouli Sastry**  *chandramouli.sastry@gmail.com*
*Dalhousie University*
*Vector Institute*
*Borealis AI*

**Andreas M. Lehrmann**  *andreas.lehrmann@gmail.com*
*Borealis AI*

**Marcus Brubaker**  *mbrubake@yorku.ca*
*York University*
*Vector Institute*
*Borealis AI*

**Alexander Radovic**  *a.radovic@gmail.com*
*Borealis AI*

**Reviewed on OpenReview:** *https://openreview.net/forum?id=LnjclqBl8R*

## Abstract

Normalizing flows model a complex target distribution in terms of a bijective transform operating on a simple base distribution. As such, they enable tractable computation of a number of important statistical quantities, particularly likelihoods and samples. Despite these appealing properties, the computation of more complex inference tasks, such as the cumulative distribution function (CDF) over a complex region (e.g., a polytope) remains challenging. Traditional CDF approximations using Monte-Carlo techniques are unbiased but have unbounded variance and low sample efficiency. Instead, we build upon the diffeomorphic properties of normalizing flows and leverage the divergence theorem to estimate the CDF over a closed region in target space in terms of the flux across its *boundary*, as induced by the normalizing flow. We describe both deterministic and stochastic instances of this estimator: while the deterministic variant iteratively improves the estimate by strategically subdividing the boundary, the stochastic variant provides unbiased estimates. Our experiments on popular flow architectures and UCI benchmark datasets show a marked improvement in sample efficiency as compared to traditional estimators.

## 1 Introduction

Normalizing Flows (Kobyzev et al., 2020; Papamakarios et al., 2021) are a form of generative model which constructs tractable probability distributions through invertible, differentiable transformations, *i.e.*, diffeomorphisms. They admit both efficient and exact density evaluation and sampling and, as valid probability distributions, theoretically support a range of other probabilistic inference tasks. However, some inference tasks remain computationally challenging. Here we consider the task of computing cumulative densities over arbitrary closed regions of distributions represented by normalizing flows. Cumulative densities are a fundamental statistical measure that answer the question: what is the probability of a sample in this range of values? There are a number of different extensions of the traditional one-dimensional definition of a CDF to higher dimensions. Here we follow conventions in related works (Botev (2017); Cunningham et al. (2013)) which define the CDF in higher dimensions as an integral over a closed region. Example applications are particularly common in the domain of uncertainty estimation (Liu et al., 2019; Mazaheri et al., 2020; Abdar et al., 2021), the evaluation of risk in financial settings (Richardson et al., 1997; Mehta et al., 2012), and

actuarial analysis in general (Meneguzzo & Vecchiato, 2004). Other recent examples include distributional reinforcement learning (Sun et al., 2021), few-shot learning (Ridgeway & Mozer, 2018), and alternative methods for training normalizing flows (Dai & Seljak, 2021).

Despite the importance of cumulative densities, current techniques for computing them with normalizing flows are restricted to traditional Monte-Carlo based estimators which, while unbiased, can have unbounded variance and poor computational efficiency. Further, such approaches fail to exploit the inherent structure of normalizing flows, namely their construction as a diffeomorphic transformation.

In this paper we describe a novel estimator that exploits the unique characteristics of normalizing flows to efficiently estimate the cumulative density in closed regions. To do this we exploit the homeomorphic property of normalizing flow transforms to relate the cumulative density in the target space to volume in the base space. We then adapt the divergence theorem to show that we only need to consider the boundary of the region in the target space.

We analyze the resulting theoretical estimator to identify best-case performance scenarios and use this analysis to motivate an adaptive, approximate estimator. To practically realize this estimator and evaluate its performance, we explore its application to the computation of cumulative densities over convex regions.

Figure 1: **Overview.** We propose an efficient estimator to compute the cumulative density $P$ induced by a normalizing flow $f$ over a complex region $\mathbf{V}$. We show that $P$ corresponds to integrating over the boundary $\partial \mathbf{W} = g(\partial \mathbf{V})$ in a uniform base space w.r.t. a vector field $\mathbf{F}$ with constant divergence 1, which can be expressed as an integration over the boundary $\partial \mathbf{V}$ in the target space w.r.t. an equivalent vector field $\mathbf{G}$.

**Contributions.** The contributions of this paper are as follows: first, we derive a novel formulation of the cumulative density of a normalizing flow by relating it to the volume in a (uniform) base space and the boundary integrals in both base and data space based on the divergence theorem. Second, based on these relations, we describe an unbiased, stochastic estimator for cumulative densities over a closed region. Next, we analyze this estimator to build an adaptive, deterministic approximation for cumulative densities over a closed region by strategically adding points along the boundary of the region. Finally, we describe a comprehensive evaluation protocol to evaluate CDF estimators for normalizing flows as a function of region sizes, architecture families, and capacities. The code to reproduce our results, including training popular normalizing flow architectures, approximating cumulative densities with the proposed adaptive boundary estimator, and other baseline methods is publicly available.[1]

## 2 Related Work

Our proposed method is principally related to two bodies of existing work: (1) algorithms for efficient estimation of cumulative densities for popular distribution functions; and (2) modifications of popular normalizing flows architectures to allow for more efficient cumulative density estimation.

**Classic Algorithms**. Computing cumulative densities is not tractable in higher dimensions even for distributions such as the multivariate Gaussian which have closed-form density functions. Consequently, approximations to volume integrals/cumulative density functions have been an important research problem: for example, Genz & Bretz (2009) discuss methods for approximating integrals over multivariate Gaussian and t-distributions. Botev & l'Ecuyer (2015) and Botev (2017) propose a tilting method for i.i.d. sampling and approximating cumulative densities over convex polytopes in multivariate Gaussian and t-distributions respectively. Approximating integrals over 1D distributions is also challenging for certain distributions;

---

[1] https://github.com/BorealisAI/nflow-cdf-approximations

see Lange (1999) for a discussion on quadrature methods and chapters 4-6 of Press et al. (1988) for a discussion on numerical methods for estimating 1D integrals. Cunningham et al. (2013) extensively study Expectation-Propagation (Minka, 2001) as an approximate integration method for computing cumulative densities over polyhedral or hyperrectangular regions in Gaussian space. In cases where the base space of a normalizing flow is Gaussian, one could consider applying this method to the region in Gaussian space corresponding to the region $\mathbf{V}$ in the target space; however, it is not easy to ensure that polyhedral regions in target space remain polyhedral in base space without restricting the expressive power of the flow transform.

**Normalizing Flows**. Cundy & Ermon (2020) consider the general problem of estimating integrals and propose an approximate inference procedure that allows explicit control of the bias-variance tradeoff. The method involves first creating partitions of the base uniform space of the normalizing flow and then training a separate variational approximator for each partition. When one partition is used for the entire region, the algorithm reduces to the standard variational approximation resulting in a biased low-variance estimate – however, for a good estimate, the number of partitions have to be increased and the algorithm reduces to Monte-Carlo sampling. Furthermore, it is not easy to determine the partitioning strategy to arrive at a good estimate with as few samples as possible. Similar to our problem setting, Liang et al. (2020) consider the problem of approximating range densities in a tabular database using auto-regressive models. In order to efficiently marginalize over data dimensions, they train the model by randomly masking some data dimensions, which is referred to as variable-skipping. At inference time, they only need to sample the variables whose ranges are specified to form the estimate. Even though the training objective is indeed the maximum-likelihood objective, the estimate obtained from the model trained with variable-skipping is only an approximate value of the range densities.

**Copulas.** Introduced in Sklar (1959), copulas are multivariate cumulative density functions which can be used to define multivariate distributions in terms of their univariate marginals. The univariate marginals and the dependence between the variables are defined separately, allowing one to tractably compute multivariate axis-aligned CDFs. For example, Chilinski & Silva (2020) suggest an alternative to normalizing flows based on copula theory wherein CDF estimates are available through a simple forward pass; nonetheless, this approach has some limitations – most notably, sampling from the learned distribution is not tractable. In contrast, we focus on computing CDFs over flexibly shaped regions with normalizing flows, a broad and flexible class of distributions.

## 3 Background

Before we describe our approach we provide a brief review of normalizing flows and traditional sampling-based methods of estimating their cumulative densities.

### 3.1 Normalizing Flows

Normalizing flows transform a random variable $\mathbf{Y}$ with a simple base distribution $p_{\mathbf{Y}}(\mathbf{y})$ into a random variable $\mathbf{X}$ with a complex target distribution $p_{\mathbf{X}}(\mathbf{x})$ using a bijective mapping $f : \mathbb{R}^d \longrightarrow \mathbb{R}^d$. Base and target space are connected by the change-of-variables formula

$$p_{\mathbf{X}}(\mathbf{x}) = p_{\mathbf{Y}}(\mathbf{y}) \, \text{abs}\left(\left|\frac{\mathrm{d}g}{\mathrm{d}\mathbf{x}}\right|\right), \tag{1}$$

where $|\cdot|$ is the determinant, $\text{abs}(\cdot)$ denotes absolute value, $g$ is the inverse of $f$, and $\frac{\mathrm{d}g}{\mathrm{d}\mathbf{x}}$ is the $(d \times d)$ – Jacobian $J$ of $g$. We set $\mathbf{Y} \sim \mathcal{U}_{[0,1]^d}$, which has the important property $P(\mathbf{y} \in \mathbf{W}) = \text{vol}(\mathbf{W})$ for any compact set $\mathbf{W} \subset [0,1]^d$, *i.e.*, the cumulative density equals the enclosed volume.[2] For pre-existing flows $f'$ with non-uniform (*e.g.*, Gaussian) base distribution $p_{\mathbf{Y}'}(\mathbf{y}')$, we can simply set $f := f'(F_{\mathbf{Y}'}^{-1}(\mathbf{y}))$ to align them with the case above. Sampling from a normalizing flow involves first sampling $\mathbf{y} \sim p_{\mathbf{Y}}$ and then applying the flow transform $f$ to obtain $\mathbf{x} = f(\mathbf{y})$.

---

[2]We use $\mathcal{U}_S$ to denote the uniform distribution with support $S$.

**Objective.** Given a compact set of interest $\mathbf{V} \subset \mathbb{R}^d$ in target space, our goal will be the efficient computation of

$$P(\mathbf{x} \in \mathbf{V}) = \int_{\mathbf{V}} p_{\mathbf{X}}(\mathbf{x}) \; \mathrm{d}\mathbf{x}. \tag{2}$$

### 3.2 Monte-Carlo Estimation

If the volume $\mathbf{V}$ admits a tractable membership function, one could estimate $P(\mathbf{x} \in \mathbf{V})$ from $N$ random samples from the model as the fraction of points falling into $\mathbf{V}$; we refer to this method as the *Monte-Carlo (MC) estimate*. Similarly, if the volume $\mathbf{V}$ admits a uniformly distributed random variable $\mathbf{U} \sim \mathcal{U}_{\mathbf{V}}$, one could estimate $P(\mathbf{x} \in \mathbf{V})$ as

$$\int_{\mathbf{V}} p_{\mathbf{X}}(\mathbf{x}) \; \mathrm{d}\mathbf{x} = \int_{\mathbf{V}} p_{\mathbf{U}}(\mathbf{x}) \cdot \frac{p_{\mathbf{X}}(\mathbf{x})}{p_{\mathbf{U}}(\mathbf{x})} \; \mathrm{d}\mathbf{x} = \mathbb{E}_{\mathbf{x} \sim p_{\mathbf{U}}} \left[ \frac{p_{\mathbf{X}}(\mathbf{x})}{p_{\mathbf{U}}(\mathbf{x})} \right],$$

which we refer to as the *Importance Sampling (IS) estimate*. While both of the described estimators are unbiased, they have unbounded variances for smaller sample sizes and may require several runs to obtain accurate results (Cunningham et al., 2013).

## 4 Efficient Estimation of Cumulative Densities

Equipped with these tools, we will now describe our technique to efficiently compute the cumulative density of Eq.(2). We proceed in three steps: first, we leverage the divergence theorem to relate $P(\mathbf{x} \in \mathbf{V})$ to a flux – mathematically expressed as the surface integral of a given vector field and a measure of *how much* of the vector field passes through a closed surface – through the *transformed* boundary $g(\partial \mathbf{V})$ of $\mathbf{V}$ in base space (Section 4.1). Then, we introduce an equivalent flux through the *untransformed* boundary $\partial \mathbf{V}$ of $\mathbf{V}$ to enable direct integration in target space (Section 4.2). Finally, we propose an iterative protocol that enables fine-grained control over the quality of the approximation (Section 4.3).

### 4.1 Cumulative Density as Boundary Flux

We will first show that computing the cumulative density over $\mathbf{V}$ in target space is equivalent to computing the volume of $\mathbf{W} := g(\mathbf{V})$ in base space.

**Lemma 1.** Let $\mathbf{V} \subset \mathbb{R}^d$ be compact and $\mathbf{W} := g(\mathbf{V})$. Then $P(\mathbf{x} \in \mathbf{V}) = \mathrm{vol}(\mathbf{W})$.

*Proof.* Noting that $p(\mathbf{x}) = p(\mathbf{y}) \, \mathrm{abs}\left(\left|\frac{\mathrm{d}g}{\mathrm{d}\mathbf{x}}\right|\right)$ and $\mathrm{d}\mathbf{y} = \mathrm{abs}\left(\left|\frac{\mathrm{d}g}{\mathrm{d}\mathbf{x}}\right|\right) \mathrm{d}\mathbf{x}$, we have

$$\begin{aligned} P(\mathbf{x} \in \mathbf{V}) = \int_{\mathbf{V}} p(\mathbf{x}) \; \mathrm{d}\mathbf{x} &= \int_{\mathbf{V}} p(\mathbf{y}) \, \mathrm{abs}\left(\left|\frac{\mathrm{d}g}{\mathrm{d}\mathbf{x}}\right|\right) \; \mathrm{d}\mathbf{x} \\ &= \int_{\mathbf{W}} p(\mathbf{y}) \; \mathrm{d}\mathbf{y} = \int_{\mathbf{W}} \mathrm{d}\mathbf{y} = \mathrm{vol}(\mathbf{W}) \; \square \end{aligned} \tag{3}$$

We can control the properties of $\mathbf{W}$ through $f$:

**Definition 1 (Diffeomorphism).** A differentiable bijection $f$ on $\mathbb{R}^d$ is called a diffeomorphism, if it has a differentiable inverse $g = f^{-1}$.

Importantly, diffeomorphisms map points on the boundary to points on the boundary and points in the interior to points in the interior (Armstrong, 2013), implying $\partial \mathbf{W} = g(\partial \mathbf{V})$ for any diffeomorphic flow $f$. This includes popular flow architectures like Glow Kingma & Dhariwal (2018), MAF Papamakarios et al. (2017), and FFJORD Grathwohl et al. (2019) and ensures that $\mathbf{W}$ meets the requirements for the following theorem relating volume integrals to surface integrals.

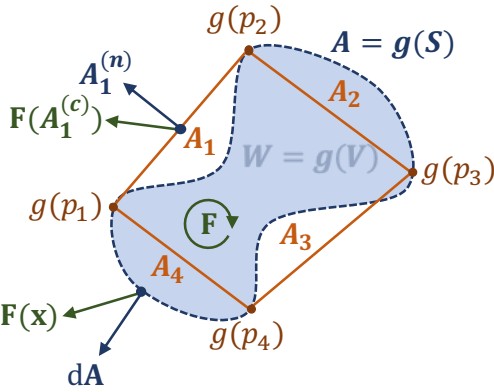

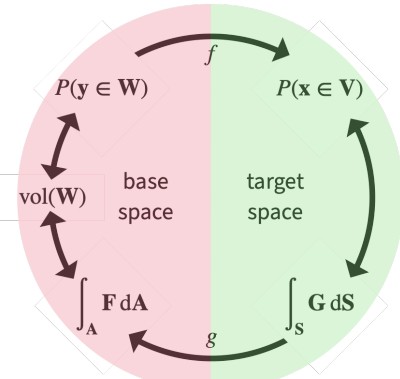

Figure 2: **Estimation in Base Space**. A piecewise linear boundary **S** defined by points $\{p_i\}_i$ in target space has a complex shape **A** in base space. The cumulative density enclosed by **S** is equivalent to the boundary flux of **F** through **A** (Eq.(5)). As an approximation, we can instead compute the boundary flux of **F** through the piecewise linear boundary $\{A_i\}_i$ defined by $\{g(p_i)\}_i$ (Eq.(6)).

Figure 3: **Equivalencies.** We show equivalent expressions for the cumulative density of interest $P(\mathbf{x} \in \mathbf{V})$ and their associated spaces. The proposed estimator is motivated by a counter-clockwise chain of arguments, ultimately leveraging $\int_{\mathbf{S}} \mathbf{G} \, d\mathbf{S}$.

**Theorem 1 (Divergence Theorem).** Let $\mathbf{T} \subset \mathbb{R}^d$ be compact with piecewise smooth boundary $\partial \mathbf{T}$. Given a vector field $\mathbf{B}$, the volume integral of the divergence $\nabla \cdot \mathbf{B}$ over $\mathbf{T}$ and the surface integral of $\mathbf{B}$ over $\partial \mathbf{T}$ are related by

$$\int_{\mathbf{T}} (\nabla \cdot \mathbf{B}) \, d\mathbf{T} = \int_{\partial \mathbf{T}} \mathbf{B} \cdot \mathbf{n} \, d(\partial \mathbf{T}), \tag{4}$$

where $\mathbf{n}$ is the outward-facing unit normal.

*Proof.* Wade (2017) $\square$

Setting $\mathbf{A} := \partial \mathbf{W}$, an application of the divergence theorem using the vector field $\mathbf{F}(\mathbf{x}) = \frac{\mathbf{x}}{d}$ yields $\nabla \cdot \mathbf{F} = 1$ and thus

$$\text{vol}(\mathbf{W}) = \int_{\mathbf{W}} d\mathbf{W} = \int_{\mathbf{W}} \nabla \cdot \mathbf{F} \, d\mathbf{W} = \int_{\mathbf{A}} \mathbf{F} \cdot \mathbf{n} \, d\mathbf{A}, \tag{5}$$

*i.e.*, we can compute the volume of $\mathbf{W}$ in terms of a flux through the boundary of $\mathbf{W}$.[3] Now, consider that the closed region $\mathbf{V}$ is given by a simplicial polytope and note that a simplicial polytope can be constructed with arbitrary accuracy for any closed region of a $d-$dimensional space whose boundary is defined by a $c < d-$dimensional manifold. The boundary of such a polytope is defined by a set of $(d-1)$-simplices; for example, the boundary of a simplicial polytope in 2D is a set of 1-simplices (*i.e.*, line segments). We can approximate the boundary $\mathbf{A}$ by flowing the vertices of the $(d-1)$-simplices defining the boundary of $\mathbf{V}$ through $g$. Denoting the resulting set of simplices by $\{A_i\}_i$, we have

$$\text{vol}(\mathbf{W}) \approx \sum_{\mathbf{A}_i} \mathbf{F}(\mathbf{A}_i^{(c)}) \cdot \mathbf{A}_i^{(n)}, \tag{6}$$

where $\mathbf{A}_i$ is the $i$-th simplex, $\mathbf{A}_i^{(c)}$ its centroid, and $\mathbf{A}_i^{(n)}$ its outward-facing normal. The approximation error is rooted in the assumption that the true boundary patch in $\mathbf{A}$ corresponding to a transformed simplex $\mathbf{A}_i$ is linear. The larger the set of simplices approximating the boundary $\mathbf{A}$ the better the estimate; see Fig. 2 for a summary of this section.

## 4.2 Stochastic Estimation in Target Space

One drawback of the method described in Eq.(6) is that it requires first constructing the set of simplices $\{A_i\}_i$ in base space before we can apply the divergence theorem. In particular, we would need to recompute

---

[3]We note that $\mathbf{F}$ is not unique and other choices are possible.

the centroids $\mathbf{A}_i^{(c)}$ and normals $\mathbf{A}_i^{(n)}$ every time $\{\mathbf{A}_i\}_i$ changes. Preferable is a direct integration over the boundary $\mathbf{S} := \partial\mathbf{V}$ in target space using an appropriate transformation of $\mathbf{F}$.

**Lemma 2.** Let $d\mathbf{A}$ and $d\mathbf{S}$ be the area vectors in base space and target space, respectively. Then we have $\int_{\mathbf{A}} \mathbf{F} \cdot d\mathbf{A} = \int_{\mathbf{S}} \mathbf{G} \cdot d\mathbf{S}$, with $\mathbf{G} := |J|J^{-1}\mathbf{F}$ and $J := \nabla g$.

*Proof.* Consider a simplex with vertices $p_i$ ($i = 1, 2, \ldots, d$) in target space and its corresponding points $g(p_i)$ defining a simplex $\Delta\mathbf{A} \in \{\mathbf{A}_i\}_i$ in base space. $g$ can be approximated with a Taylor expansion

$$g(\mathbf{x} + \epsilon) = g(\mathbf{x}) + J\epsilon + O(||\epsilon||_2^2), \tag{7}$$

where $J$ is the $(d \times d)$ – Jacobian at the centroid of the simplex (as the transformation $g$ preserves input size). With $\mathbf{F} = \mathbf{F}(\Delta\mathbf{A}^{(c)}) \in \mathbb{R}^d$ denoting the field at the centroid, we can write out $\mathbf{F} \cdot \Delta\mathbf{A}^{(n)}$ as

$$\mathbf{F} \cdot \Delta\mathbf{A}^{(n)} = \frac{1}{(d-1)!} \left| \left[ \begin{array}{c} \mathbf{F} \\ J(p_2 - p_1) + 2O(||\epsilon||_2^2) \\ J(p_3 - p_1) + 2O(||\epsilon||_2^2) \\ \vdots \\ J(p_d - p_1) + 2O(||\epsilon||_2^2) \end{array} \right] \right|. \tag{8}$$

See Appendix A for a general discussion on computing surface normals. In the limit $\Delta\mathbf{A} \to 0$, we have $\Delta\mathbf{A}^{(n)} \to d\mathbf{A}$ and $||\epsilon||_2^2 \to 0$:

$$\mathbf{F} \cdot d\mathbf{A} = \frac{1}{(d-1)!} \left| J \left[ \begin{array}{c} J^{-1}\mathbf{F} \\ p_2 - p_1 \\ p_3 - p_1 \\ \vdots \\ p_d - p_1 \end{array} \right] \right| = \mathbf{G} \cdot d\mathbf{S}, \tag{9}$$

where $\mathbf{G} = |J|J^{-1}\mathbf{F}$ $\square$

For notational convenience we assume $|J| > 0$ so that outward-facing surface normals in base space remain outward-facing in target space; the general case requires straightforward tracking of their signs. Interestingly, $\mathbf{G}$ has a tractable divergence, paving the way for an application of the divergence theorem in target space.

**Lemma 3.** With $\mathbf{G}$ defined as in Lemma 2, we have $\nabla \cdot \mathbf{G} = |J|$.

*Proof.*

$$\begin{aligned} \nabla \cdot \mathbf{G} &= \nabla \cdot (\text{adj}(J)\mathbf{F}) \\ &= (\nabla \cdot \text{adj}(J))\mathbf{F} + \text{tr}(\text{adj}(J)\nabla_x\{\mathbf{F}\}) \\ &= d^{-1}\text{tr}(\text{adj}(J)J) + (\nabla \cdot \text{adj}(J))\mathbf{F} \\ &= |J| + (\nabla \cdot \text{adj}(J))\mathbf{F} \\ &= |J|, \end{aligned} \tag{10}$$

because $\text{tr}(\text{adj}(J)J) = d|J|$. Furthermore, Evans (2010) shows $\nabla \cdot \text{adj}(J) = 0$ for any $C^2$-map $\square$

Lemma 3 allows us to close the loop and tie the field $\mathbf{G}$ back to our original objective of computing $P(\mathbf{x} \in \mathbf{V})$. Indeed, an application of Eq.(4) and Eq.(1) shows

$$\int_{\mathbf{S}} \mathbf{G} \cdot d\mathbf{S} = \int_{\mathbf{V}} |J| \, d\mathbf{V} = \int_{\mathbf{V}} p(\mathbf{x}) \, d\mathbf{V} = P(\mathbf{x} \in \mathbf{V}). \tag{11}$$

Fig. 1 shows how the flux of a learned vector field $\mathbf{G}$ across the boundary $\partial\mathbf{V}$ is related to $\text{vol}(\mathbf{W})$, which can be expressed as the flux of a vector field $\mathbf{F}$ with $\nabla \cdot \mathbf{F} = 1$ across the boundary $\partial\mathbf{W}$ – obtained by applying a learned transformation $g$. See Fig. 3 for an overview of all equivalencies derived.

### 4.2.1 Stochastic Boundary Flux Estimator

For a simplicial polytope $\mathbf{V}$ with boundary $\mathbf{S}$ given by non-overlapping simplices $\{\mathbf{S}_i\}_i$, we can leverage $\mathbf{G}$ to define a *stochastic boundary flux (BF-S)* estimator of the cumulative density $P(\mathbf{x} \in \mathbf{V})$ using points sampled from the boundary:

$$
\int_{\mathbf{S}} \mathbf{G} \cdot \mathrm{d}\mathbf{S} = \sum_i \int_{\mathbf{S}_i} \mathbf{G}(\mathbf{x}) \cdot \mathbf{S}_i^{(n)} \, \mathrm{d}\mathbf{x} = \sum_i \mathbf{S}_i^{(A)} \int_{\mathbf{S}_i} \frac{\mathbf{G}(\mathbf{x})}{\mathbf{S}_i^{(A)}} \cdot \mathbf{S}_i^{(n)} \, \mathrm{d}\mathbf{x} = \sum_i \mathbf{S}_i^{(A)} \int_{\mathbf{S}_i} \mathcal{U}_{\mathbf{S}_i}(\mathbf{x}) \cdot \mathbf{G}(\mathbf{x}) \cdot \mathbf{S}_i^{(n)} \, \mathrm{d}\mathbf{x}
$$
$$
= \sum_i \mathbf{S}_i^{(A)} \cdot \mathbb{E}_{\mathbf{x} \sim \mathcal{U}_{\mathbf{S}_i}}[\mathbf{G}(\mathbf{x}) \cdot \mathbf{S}_i^{(n)}],
\tag{12}
$$

where $\mathbf{S}_i$ is the $i$-th boundary simplex, $\mathbf{S}_i^{(A)}$ its area, and $\mathbf{S}_i^{(n)}$ its outward-facing unit normal. Alternatively, multiplying and dividing the surface integral $\int_{\mathbf{S}} \mathbf{G} \cdot \mathrm{d}\mathbf{S}$ with the total surface area $\left(\sum_i \mathbf{S}_i^{(A)}\right)$:

$$
\int_{\mathbf{S}} \mathbf{G} \cdot \mathrm{d}\mathbf{S} = \left(\sum_i \mathbf{S}_i^{(A)}\right) \cdot \mathbb{E}_{\mathbf{x} \sim \mathcal{U}_{\mathbf{S}}}\left[\mathbf{G}(\mathbf{x}) \cdot \mathbf{n}(\mathbf{x})\right],
\tag{13}
$$

where $\mathbf{n}(\mathbf{x})$ is the outward-facing unit normal at $\mathbf{x}$. Note that $\mathcal{U}_{\mathbf{S}}$ can be decomposed in terms of a categorical distribution $c_i = \mathbf{S}_i^{(A)} / \sum_j \mathbf{S}_j^{(A)}$ over the simplices $\mathbf{S}_i$ and the uniform distributions $\mathcal{U}_{\mathbf{S}_i}$. More specifically, we have $\mathcal{U}_{\mathbf{S}}(\mathbf{x}) = \sum_i c_i \cdot \mathcal{U}_{\mathbf{S}_i}(\mathbf{x})$, wherein all but one of the terms are zero for a point $\mathbf{x}$ on the surface $\mathbf{S}$ — because the simplices are non-overlapping and the point $\mathbf{x}$ belongs to exactly one of them.

## 4.3 Adaptive Estimation in Target Space

While the stochastic boundary flux estimator introduced in Section 4.2.1 is unbiased, it does not allow for a strategic placement of evaluation points in an iterative fashion. We can turn Eq.(12) into a sequential process by first approximating $\mathbb{E}_{\mathbf{x} \sim \mathcal{U}_{\mathbf{S}_i}}[\mathbf{G}(\mathbf{x}) \cdot \mathbf{S}_i^{(n)}]$ with a deterministic estimate and then performing prioritized iterative refinements.

### 4.3.1 Deterministic Boundary Flux Estimator

Given a set of boundary simplices $\{\mathbf{S}_i\}_i$, we can approximate the expected dot-product in Eq.(12) by the average dot-product over the *vertices* of $\mathbf{S}_i$. Formally, this *adaptive boundary flux (BF-A)* estimator of $P(\mathbf{x} \in \mathbf{V})$ is given by

$$
\int_{\mathbf{S}} \mathbf{G} \cdot \mathrm{d}\mathbf{S} \approx \sum_i \mathbf{S}_i^{(A)} \cdot \overline{\mathbf{G}}(\mathbf{S}_i) \cdot \mathbf{S}_i^{(n)},
\tag{14}
$$

where $\overline{\mathbf{G}}(\mathbf{S}_i)$ is the average $\mathbf{G}$-field over all vertices of $\mathbf{S}_i$. While Eq.(14) is a biased estimate of the cumulative density $P(\mathbf{x} \in \mathbf{V})$, its deterministic nature and inherent structure increase sample efficiency (*e.g.*, prudent placement, reuseable computations) and enable strategic refinements to arbitrary precision.

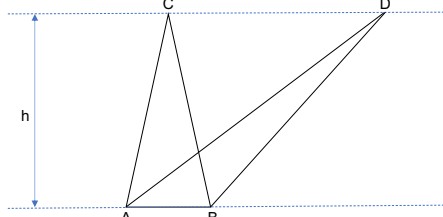

### 4.3.2 Iterative Refinement

The quality of the approximation in Eq.(14) can be controlled via iterative refinement of the initial set of boundary simplices $\{\mathbf{S}_i\}_i$. Each refinement step consists of a splitting operation that first introduces one new vertex at the midpoint of a boundary line segment[4] and then splits all simplices sharing that edge. The key component of this splitting process is a priority queue $Q$ managing the next boundary line segment $l$ to split, where $l$ is the longest edge of the simplex with highest priority according to a priority function Pr.

Figure 4: **Splitting Criterion (Total Edge Length).** For two simplices (*e.g.*, triangles ABC and ABD) with equal area, the uncertainty about the dot-product within a simplex grows with a point's distance to its closest vertex, motivating the use of total edge length as a splitting criterion.

---

[4]The $(d-1)$-simplices $\{\mathbf{S}_i\}_i$ forming the boundary $\mathbf{S}$ each consist of $\binom{d}{2}$ line segments.

Pr is designed such that it gives high priority to simplices which *probably* have high error. As a general rule, the larger the simplex the larger the variation in the dot-products, and hence the larger the estimation bias. Although the area $\mathbf{S}_i^{(A)}$ of a simplex squarely fits the definition of size, it can be a misleading measure that should not be used in isolation — see Fig. 4 for an illustrated explanation. Therefore, we also use the total edge length as a measure of size. While this approach works well in lower dimensions, we saw marked improvements in terms of sample efficiency in higher dimensions after complementing these two criteria with the standard deviation of the dot-products at the vertices. Taken together, we define the priority of the $i$-th simplex $\mathbf{S}_i$ as

$$\Pr(\mathbf{S}_i) = \mathbf{S}_i^{(A)} \times (\sigma_d(\mathbf{S}_i) + \epsilon) \times \sum_{e_j \in \mathbf{S}_i} ||e_j||_2^2, \tag{15}$$

where $e_j$ is the $j$-th edge (1-simplex) constituting the simplex $\mathbf{S}_i$, $\sigma_d(\mathbf{S}_i)$ is the standard deviation of the dot-products – computed with the unit surface normal – across the simplex $\mathbf{S}_i$, and $\epsilon$ is a small number to guard against assigning a zero priority to large simplices whose dot-products at the vertices are all zero. See Appendix B for a summary of the entire splitting process in pseudo code.

### 4.4 Analysis

#### 4.4.1 Best Case Scenarios

The integral of $\mathbf{G}$ over a simplex $\mathbf{S}_i$ can be computed exactly if the flow transformation $g$ is: (a) linear over the simplex $\mathbf{S}_i$; or (b) $\mathbf{G} \cdot \mathbf{S}_i^{(n)}$ is constant for all $\mathbf{x} \in \mathbf{S}_i$. The latter is equivalent to $\mathbf{F} \cdot d\mathbf{A}$ being constant, such as for hyperspheres. In both cases, we have $\overline{\mathbf{G}}(\mathbf{S}_i) = \mathbb{E}_{\mathbf{x} \sim \mathcal{U}_{\mathbf{S}_i}}[\mathbf{G}(\mathbf{x})]$. In practice, it is rarely possible to realize these best cases, however, if the dot-product is locally-linear over the simplex, our biased estimate is a good approximation to the exact integral. For example, observe that the field lines of $\mathbf{G}$ vary across the boundary $\partial \mathbf{V}$ in Fig. 1. The true value of $\mathbb{E}_{\mathbf{x} \sim \mathcal{U}_{\mathbf{S}_i}}[\mathbf{G}(\mathbf{x})]$ thus differs from its approximation $\overline{\mathbf{G}}(\mathbf{S}_i)$ – a gap we aim to minimize with the adaptive splitting protocol described above.

#### 4.4.2 Runtime Complexity & Implementation Considerations

For a given simplicial polytope, the algorithm first computes the surface normals for each of the boundary simplices, which requires computing $d$ determinants of $(d-1) \times (d-1)$ matrices for each boundary simplex. The total complexity of this *initialization* step is thus $\mathcal{O}(kd^4)$, where $k$ is the number of boundary simplices. Note that the surface normals do not need to be recomputed after splitting edges. Furthermore, if regions are defined as a set of half-spaces (*i.e.*, linear inequalities), the surface normals are immediately available and do not require the quartic initialization step. Following this initialization, the manipulation of the priority queue with every split is logarithmic in the number of boundary simplices.

When computing $\mathbf{G}$, one has to first flow the input to the uniform distribution and then compute the Jacobian-vector product of the backward-pass; in order to avoid computing the Jacobian-vector product, we can first compute the vector-Jacobian product $\mathbf{S}_i^{(n)\top} J^{-1}$ of the backward-pass followed by a dot-product with $|J|\mathbf{F}$ to obtain $\mathbf{S}_i^{(n)\top}|J|J^{-1}\mathbf{F}$. This avoids an extra backward-pass in PyTorch, which does not support forward-mode auto-differentiation of Jacobian-vector-products. In fact, the derivative of $\mathbf{S}_i^{(n)\top}|J|J^{-1}\mathbf{F}$ with respect to $\mathbf{S}_i^{(n)\top}$ yields exactly $\mathbf{G}$.

## 5 Experiments

We will now evaluate the proposed adaptive boundary flux (BF-A) estimator against traditional sampling-based estimators like Monte-Carlo (MC) and Importance Sampling (IS). As the accuracies of the CDF estimates of the baseline estimators are mainly a function of sample size (Cunningham et al., 2013) we compare all methods based on sample budgets.

For the purpose of evaluation, we train normalizing flows on $d$-dimensional ($d \in \{2, 3, 4, 5\}$) data derived from 4 tabular datasets open sourced as part of the UCI Machine Learning Repository (Dua & Graff, 2017) and preprocessed as in Papamakarios et al. (2017): Power, Gas, Hepmass, and Miniboone. We construct

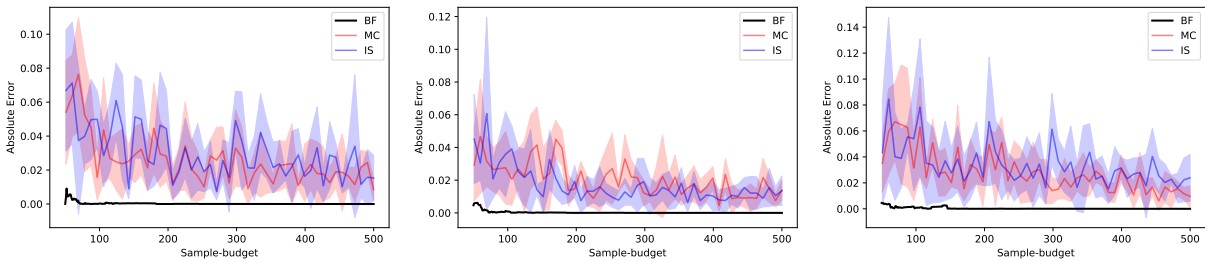

(a) Glow (Kingma & Dhariwal, 2018) (b) MAF (Papamakarios et al., 2017) (c) FFJORD (Grathwohl et al., 2019)

Figure 5: **Sample Efficiency.** We highlight the sample efficiency of the proposed adaptive boundary flux (BF-A) estimator as compared to Monte-Carlo (MC) and Importance Sampling (IS) estimators. We show absolute estimation errors for 3 flow architectures: two discrete normalizing flows (Glow and MAF) — composed of 7 flow-layers with 64 hidden units in their coupling nets — and one continuous normalizing flow (FFJORD) — parameterized by a 2-layer MLP with 64 hidden units in each layer. All models were trained on a 2D slice of the Power dataset, with volumes $\mathbf{V}$ given by convex hulls constructed from points sampled on the surface of spheres with radius 1.0, centered at points sampled from the flow.

normalizing flows from the following three architecture families: Glow (Kingma & Dhariwal, 2018), Masked Autoregressive Flow (MAF) (Papamakarios et al., 2017), and FFJORD (Grathwohl et al., 2019). For every pair (dataset, $d$), we obtain 2 random $d$-dimensional slices of the dataset over which we train the normalizing flows. Additionally, we also consider how the capacities of these models affect the performance: for the two discrete flows (Glow and MAF), we vary the number of hidden dimensions in the coupling networks and the number of flow transforms stacked together; for FFJORD, we instead vary the number of hidden dimensions in each layer of the MLP parameterizing the ODE. For constructing discrete flows, we choose 3, 5 or 7 flow layers and construct coupling layers with 16, 32 or 64 hidden units. While one Glow layer corresponds to a sequence of (ActNorm) – (Glow Coupling) – (Invertible $1 \times 1$) transformations, one MAF layer corresponds to a sequence of (ActNorm) – (MAF Coupling) transformations. For continuous flows, we parameterize the neural ODE with 2 hidden layers, each consisting of 16, 32 or 64 hidden units. In summary, we obtain a total of 168 flow models for every $d \in \{2, 3, 4, 5\}$: 9 each from the 2 discrete flow families and 3 from the continuous flow, trained on 2 slices derived from each of the 4 datasets. We refer to Appendix C for additional training details and box-plots of training log-likelihoods.

**Generating Simplicial Polytopes.** The crucial component in ensuring a fair and unbiased evaluation of the estimators is in uniformly sampling simplicial polytopes $\mathbf{V}$ in target space such that they are not biased towards any specific regions or cumulative densities. In order to actually *sample* a simplicial polytope, we construct a convex hull around a fixed number of points in target space. To ensure that convex hulls of varying sizes are equally explored, we sample these points from the boundary of a sphere centered at a flow sample and control the size of the convex hull by controlling the radius of the sphere. We observed that the inclusion of very small convex hulls can lead to inconsistent results across runs due to large relative errors of the sampling-based estimators (MC, IS). Thus we only accepted convex hulls with CDFs larger than 0.01 for fairness and to improve stability and reproducibility between runs. While the number of hull points affects the average size and cumulative density of the enclosed volume, it does not markedly change the average size of the simplices as defined in Section 4.3.2 and we therefore do not consider the number of hull points as an evaluation axis.

In our evaluation we consider 20 points to construct the hull, trading off lower cumulative densities against linear approximation of the spherical boundary.[5] Furthermore, we consider spheres with radius 0.5, 0.75, and 1.0, amounting to a coverage radius of up to 1 standard deviation for our normalized data. Finally, in order

---

[5]Note that all 20 points lie on the boundary of the hull and count against the sample budget of the BF-A algorithm.

| Estimator | Absolute Error | Relative Error |
|---|---|---|
| IS | 0.00572±0.01585 | 0.03000±0.04632 |
| MC | 0.00381±0.00381 | 0.04864±0.05178 |
| BF-S | 0.01705±0.04175 | 0.19032±0.53439 |
| BF-A (**ours**) | **0.00152±0.00554** | **0.01822±0.06626** |

Table 1: **Quantitative Evaluation (Aggregate)**. We show the absolute and relative errors of each estimator for a sample budget of 4,000 points, averaged across all evaluation axes. The proposed BF-A algorithm outperforms the IS and MC estimators in terms of both absolute and relative errors ($p < 0.001$ in a one-sided U-Test).

to evaluate a normalizing flow model, we construct 5 convex hulls for each choice of the radius. Overall, we evaluate each CDF estimator over a total of 10,080 hulls.[6]

**Evaluation.** For each convex hull we compute a reference cumulative density using $2M$ importance samples and evaluate all estimators against this gold standard based on their predicted cumulative density using a fixed sample budget $K$. Unlike our proposed adaptive boundary flux (BF-A) estimator, which is deterministic, the Monte-Carlo (MC) and Importance Sampling (IS) estimators are stochastic. Thus, in order to accurately capture the variance of these estimators, we collect results across 5 runs for each hull. We then evaluate the estimators in terms of both relative and absolute errors. Table 1 shows the aggregate results across all 10,080 configurations with a sample budget of 4,000 points. In the remainder of this section, we will unroll these results and analyze our estimator along individual evaluation axes.

**Discussion.** On average, our BF-A estimator obtains approximately $3.8\times$ and $2.5\times$ lower absolute errors than IS and MC, respectively, for a sample budget of 4,000 points (Table 1); likewise, we obtain $1.7\times$ and $2.7\times$ lower relative errors than IS and MC, respectively. For the specific case of 2D flows, we report relative errors that are better than IS by $11.1\times$ and MC by $21.2\times$ (Table 2a); also see Fig. 5 for a qualitative comparison of the estimators in 2D in terms of the absolute deviation from the true cumulative density. We note that in all 3 cases the proposed algorithm converges to the true estimate well within 500 samples, while the stochastic estimates of the MC and IS estimators exhibit high variance. This pattern is consistent across our experiments with 2D flows and we find that our algorithm outperforms IS and MC by $30\times$ and $55\times$, respectively, for a sample budget of just 500 points. Conversely, the MC and IS algorithms would require ~50k samples on average to match the relative errors of BF-A with just 4,000 samples in 2D. Averaged across all flows and dimensions (2D‑5D), we find that the baseline estimators would require $9.6\times$ as many samples to match the relative error rates of BF-A. If we also account for per-sample running time (Appendix D; Table 5), we find that the baseline estimators can process slightly more samples in the same time that would be needed by BF-A to form an estimate with 4,000 samples. However, they would still require $6.4\times$ more time on average to match the relative error rates of BF-A. Nevertheless, absolute runtime depends on a number of confounding factors – such as programming language, dedicated data structures, compute device, and low-level optimizations – making a fair comparison difficult. We view runtime optimization as an orthogonal research direction that can further capitalize on faster convergence rates and better sample efficiencies of CDF estimators.

A comparison across different hull radii is shown in Table 2b. We note that, while the absolute errors of all estimators increase with increasing hull-sizes, the relative errors of MC decrease with increasing hull sizes, while those of IS and BF-A increase with increasing hull sizes; nonetheless, BF-A outperforms IS and MC in terms of both absolute and relative errors for the explored hull sizes. Additional quantitative evaluations can be found in the Appendix: in terms of different capacities (Appendix D; Tables 3-4b), the observed trends are in line with the analysis above, both for discrete and continuous flows. We also note that varying the width of the coupling transform (in discrete flows) does not affect the observed trends as much as the depth of the flow transform. Finally, we observe similar trends across varying flow architectures as well (Appendix D; Table 4a); also see Fig. 8 in Appendix D for a visual summary of all quantitative evaluations.

---

[6]10,080 hulls = 168 $\frac{\text{models}}{\text{dim}}$ · 4 dim · 5 $\frac{\text{hulls}}{\text{models·radii}}$ · 3 radii

| Dim | Estimator | Absolute Error | Relative Error |
|---|---|---|---|
| 2 | IS | 0.00737±0.01981 | 0.01620±0.02459 |
|   | MC | 0.00482±0.00434 | 0.03101±0.03366 |
|   | BF-A (**ours**) | **0.00071±0.00281** | **0.00146±0.00308** |
| 3 | IS | 0.00528±0.01564 | 0.02543±0.03618 |
|   | MC | 0.00363±0.00381 | 0.05247±0.05202 |
|   | BF-A (**ours**) | **0.00134±0.00584** | **0.00710±0.03254** |
| 4 | IS | 0.00405±0.00818 | 0.04495±0.05404 |
|   | MC | 0.00274±0.00251 | 0.06576±0.06162 |
|   | BF-A (**ours**) | **0.00237±0.00780** | **0.03209±0.08405** |
| 5 | IS | 0.00376±0.00640 | **0.07233±0.08153** |
|   | MC | **0.00244±0.00218** | 0.07340±0.06483 |
|   | BF-A (**ours**) | 0.00380±0.00658 | 0.07796±0.14184 |

(a) **Dimension-wise**

| Radius | Estimator | Absolute Error | Relative Error |
|---|---|---|---|
| 0.5 | IS | 0.00338±0.01066 | 0.01895±0.02430 |
|   | MC | 0.00336±0.00336 | 0.05670±0.05612 |
|   | BF-A (**ours**) | **0.00073±0.00287** | **0.01016±0.04500** |
| 0.75 | IS | 0.00480±0.01327 | 0.02771±0.04061 |
|   | MC | 0.00365±0.00350 | 0.04896±0.05205 |
|   | BF-A (**ours**) | **0.00126±0.00460** | **0.01666±0.05429** |
| 1.0 | IS | 0.00808±0.01998 | 0.03941±0.05871 |
|   | MC | 0.00425±0.00428 | 0.04292±0.04761 |
|   | BF-A (**ours**) | **0.00229±0.00728** | **0.02500±0.08450** |

(b) **Size-wise**

Table 2: **Quantitative Evaluation**. We show the absolute and relative errors of each of the methods for a sample budget of 4,000 points, averaged across all configurations separately for (a) each dimensionality and (b) each radius of the hull. BF-A outperforms both IS and MC on average for flows < 5D and the explored hull sizes ($p < 0.001$ in a one-sided U-Test for each evaluation block).

Our qualitative and quantitative evaluations show a marked improvement of the proposed method over traditional sampling-based estimators. BF-A benefits from our novel formulation of cumulative densities as boundary fluxes as well as from the sample efficiency enabled by the priority function Pr. We note that BF-S, the stochastic realization of the boundary flux estimator (Eq.(13)) uses random samples on the boundary to form a Monte-Carlo estimate of the CDF and accrues a higher relative error of 0.19 on average compared to the other Monte-Carlo estimators (Table 1); nonetheless, its formulation readily extends itself to an adaptive refinement in $(d-1)$ – dimensional space, which we use in developing the BF-A estimator. Although the BF-S estimator cannot be directly applied to obtain sample-efficient and accurate CDF estimates, it could form the basis of stochastic-adaptive estimators in the future. Intuitively, the estimation error of the BF-A estimator can be traced back to how much the dot-product $\mathbf{G} \cdot \mathbf{n}$ can change for small changes in the input $\mathbf{x}$; in fact, our motivation behind evaluating the estimators as a function of growing dimensions, growing hull-sizes, and growing model capacities is targeted towards testing how well our estimator can *adapt* to increasing complexity. We observe that while our method continues to outperform the sampling-based estimators with growing hull sizes (Table 2b), model capacities (Appendix D; Table 3 and Table 4b), varying flow architectures (Appendix D; Table 4a), and data dimensions up to and including 4D, BF-A only performs comparably to the sampling-based estimators for a sample budget of 4,000 points with 5D flows (Table 2a).

**Limitations & Future Work.** A limitation of the proposed estimator is that the structural advantages and improvements over sampling-based estimators are likely to be less pronounced in higher dimensions: the CDF over a compact region is defined by a computationally challenging integration over a $d$-dimensional volume, which inherently scales poorly with dimension. Here we describe how to exploit the diffeomorphic structure of distributions defined by normalizing flows to produce a more efficient CDF estimation method and validate our approach on problems between 2D and 5D. While higher-dimensional CDFs are certainly of interest, our evaluation of equivalent sample sizes shows that even low-dimensional CDFs can be computationally demanding. Additional challenges in high-dimensional spaces, such as floating point precision, cannot be ruled out and may require dedicated solutions not covered in this work.

Future extensions of the proposed estimator could consider using higher-order information of the model, such as curvature, in defining the priority function in order to improve the scaling of the method; this may also enable the definition of a more natural stopping criterion than sample budget. Another interesting extension to the adaptive estimator would be to split a *batch* of edges together as opposed to splitting a single edge at a time. Additionally, it remains to be explored how different choices of field $\mathbf{F}$ could be used in minimizing the estimation error for a given simplicial polytope. Finally, noting that our method works better with FFJORD than MAF and Glow on average (Appendix D; Table 4a), one could also explore regularization methods that encourage smoother transformations between the target and base spaces.

## Conclusion

Cumulative density functions are as fundamentally important as probability density functions but cannot always be estimated efficiently and exactly. In this work, we propose a new type of CDF estimator, derived as an adaptation of the divergence theorem, by exploiting the diffeomorphic properties of normalizing flows. Furthermore, we demonstrate how this CDF estimator can be made more sample-efficient by reusing computations and strategically acquiring new points such that the estimate obtains a precise CDF estimate with as few points as possible. In our evaluation of this new adaptive estimator, we find that it outperforms traditional sampling-based estimators in terms of sample-efficiencies and accuracies. We believe that our proposed efficient CDF estimator will help open up new application areas with normalizing flows.

**Acknowledgements.** This work was supported by a Borealis AI Graduate Fellowship. The authors also thank Vector Institute and Sageev Oore for supporting the project with computational resources.

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

# A  Computing Surface Normals

In applying the divergence theorem, one would need to compute the surface normal vector. In the context of this paper, we compute surface normals to the $(d-1)$-dimensional simplices constituting the boundary: for example, we would need to compute the surface normal to line-segments in 2D, triangles in 3D, tetrahedrons in 4D and so on. By definition of $(d-1)$-dimensional simplices, they lie in a plane defined by $(d-1)$ linearly independent vectors; since we are computing normals to $(d-1)$-dimensional simplices lying in $\mathbb{R}^d$, the vectors spanning the plane containing the simplex and the normal vector to this plane form a full basis of $\mathbb{R}^d$. In general, one could imagine solving a system of simultaneous equations in order to determine a vector normal to a given set of vectors. However, in this specific case, since we know that we are computing a normal vector to $(d-1)$ linearly independent vectors, we can just define the normal vector to a simplex $\mathbf{z}$ defined by the vertex set $\{z_i\}_i$ in terms of the determinant:

$$\mathbf{v} = \left\| \begin{bmatrix} \mathbf{e} \\ z_2 - z_1 \\ z_3 - z_1 \\ \vdots \\ z_d - z_1 \end{bmatrix} \right\| \tag{16}$$

where, $\mathbf{e}$ is the orthonormal unit basis in $\mathbb{R}^d$. Since the determinant of a matrix containing two identical rows evaluates to 0, we can easily confirm that the dot product $\forall i \in [2,d]$, $\mathbf{v} \cdot (z_i - z_1) = 0$. Also note that at a high level, this construction resembles the vector cross-product operation. In fact, similar to vector cross products, the vector norm of the $\mathbf{v}$ is equal to the volume of the parallelotope spanned by the set of $(d-1)$ linearly independent vectors $\{z_i - z_1\}_i$. Therefore, in order to derive the surface normal vector whose magnitude is equal to the volume of the simplex $\mathbf{z}$, we would need to divide $\mathbf{v}$ with $(d-1)!$:

$$\mathbf{z}^{(n)} = \frac{1}{(d-1)!} \left\| \begin{bmatrix} \mathbf{e} \\ z_2 - z_1 \\ z_3 - z_1 \\ \vdots \\ z_d - z_1 \end{bmatrix} \right\| \tag{17}$$

Accordingly, for some vector $\mathbf{w}$, the dot-product $\mathbf{w} \cdot \mathbf{z}^{(n)}$ can be written as:

$$\mathbf{w} \cdot \mathbf{z}^{(n)} = \frac{1}{(d-1)!} \left\| \begin{bmatrix} \mathbf{w} \\ z_2 - z_1 \\ z_3 - z_1 \\ \vdots \\ z_d - z_1 \end{bmatrix} \right\| \tag{18}$$

# B   Pseudo-codes

---

**Algorithm 1** Split edges and update the volume

---

**Objects and Attributes:**

       edge: A 1 simplex object in $\mathbb{R}^d$ having the following attributes:

          – endPoints: the endPoints of the edge

          – simplices: set of $d-1$-dimensional simplices that contains this edge

       simplex: A $d-1$ simplex object in $\mathbb{R}^d$ having the following attributes:

          – edges: the set of edges contained in the simplex

          – unitSurfaceNormal: the unit surface normal of the simplex, adjusted so that it points outwards (or inwards) of the hull.

          – volume: the volume of the simplex

          – getVolumeElement: a method that will loop through the points and computes the mean dot-product.

**Input:**

       edgeToSplit: The selected edge to split according to the priority function described in Eq.(15).

       edges: Hashmap of Edges

          – Allows $\mathcal{O}(1)$ retrieval of edge objects from the endpoints. If object does not exist in the queue, a new one is created and returned.

       volume: The current estimated volume

**Output:**

       newVolume

```
 1: newVolume ← volume                                    ▷ Initialize the newVolume to be equal to the current volume
 2: for simplex in edgeToSplit.simplices do
 3:     simplex1 ← new Simplex()
 4:     simplex2 ← new Simplex()
 5:     simplex1.unitSurfaceNormal ← simplex.unitSurfaceNormal
 6:     simplex2.unitSurfaceNormal ← simplex.unitSurfaceNormal
 7:     simplex1.volume ← simplex.volume/2
 8:     simplex2.volume ← simplex.volume/2
 9:     for otherEdge in simplex.edges-{edgeToSplit} do
10:         newEdge1 ← edges.get(otherEdge.endPoint[0],edge.midPoint)
11:         newEdge2 ← edges.get(otherEdge.endPoint[1],edge.midPoint)
12:         if otherEdge.hasEndPoint(edgeToSplit.endPoint[0]) then
13:             otherEdge.add(simplex1)
14:             simplex1.edges.add(otherEdge, newEdge1, newEdge2)
15:         else if otherEdge.hasEndPoint(edgeToSplit.endPoint[1]) then
16:             otherEdge.add(simplex2)
17:             simplex2.edges.add(otherEdge, newEdge1, newEdge2)
18:         else
19:             otherEdge.add(simplex1, simplex2)
20:             newEdge1.add(simplex1, simplex2)
21:             newEdge2.add(simplex1, simplex2)
22:             simplex1.edges.add(otherEdge, newEdge1, newEdge2)
23:             simplex2.edges.add(otherEdge, newEdge1, newEdge2)
24:         otherEdge.remove(simplex)
25:     newVolume ← newVolume - simplex.getVolumeElement()
26:     newVolume ← newVolume + simplex1.getVolumeElement()
27:     newVolume ← newVolume + simplex2.getVolumeElement()
28: return newVolume
```

---

---

**Algorithm 2** getVolumeElement of Simplex Object

---

**Objects and Attributes:**

       edge: A 1 simplex object in $\mathbb{R}^d$ having the following attributes:

            – endPoints: the endPoints of the edge

            – simplices: set of $d-1$ simplices that contains this edge

       simplex: A $d-1$ simplex object in $\mathbb{R}^d$ having the following attributes:

            – edges: the set of edges contained in the simplex

            – unitSurfaceNormal: the unit surface normal of the simplex, adjusted so that it points outwards (or inwards) of the hull.

            – volume: the volume of the simplex

            – getVolumeElement: a method that will loop through the points and computes the mean dot-product.

**Input:**

       simplex: A simplex object

       flow: The normalizing flow

       D: Number of dimensions

**Output:**

        volume

1: volumeElement ← 0
2: **for** point in simplex.points() **do**
3:     volumeElement ← volumeElement + simplex.volume*flow.dotProduct(point,simplex.unitSurfaceNormal)
4: **return** volumeElement

---

## C    Training Details and Log-likelihoods

For all normalizing flows, we train the models with a batch size of 10k and stop when the log-likelihoods do not improve over 5 epochs. For the continuous flows, we used the exact divergence for computing the log-determinant. We used exp-scaling in the affine coupling layer of both MAF and Glow models – and, in order to prevent numerical overflows, we applied a tanh nonlinearity before the exp-scaling. Finally, we used softplus as our activation function for both the Neural ODE and coupling networks. From Fig. 6 and Fig. 7, we observe both the Continuous and Discrete flows obtain similar log-likelihoods and are able to fit the training data well.

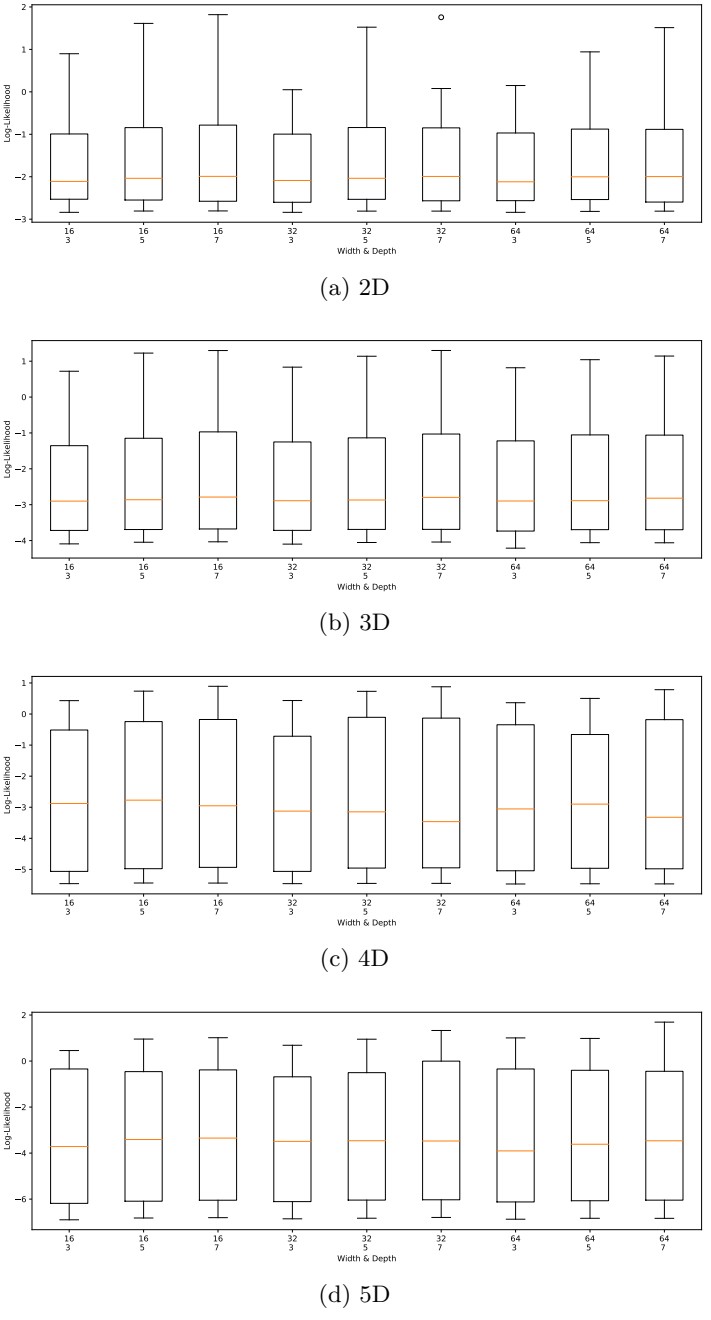

(a) 2D

(b) 3D

(c) 4D

(d) 5D

Figure 6: **Log-Likelihoods (Discrete Flows).**

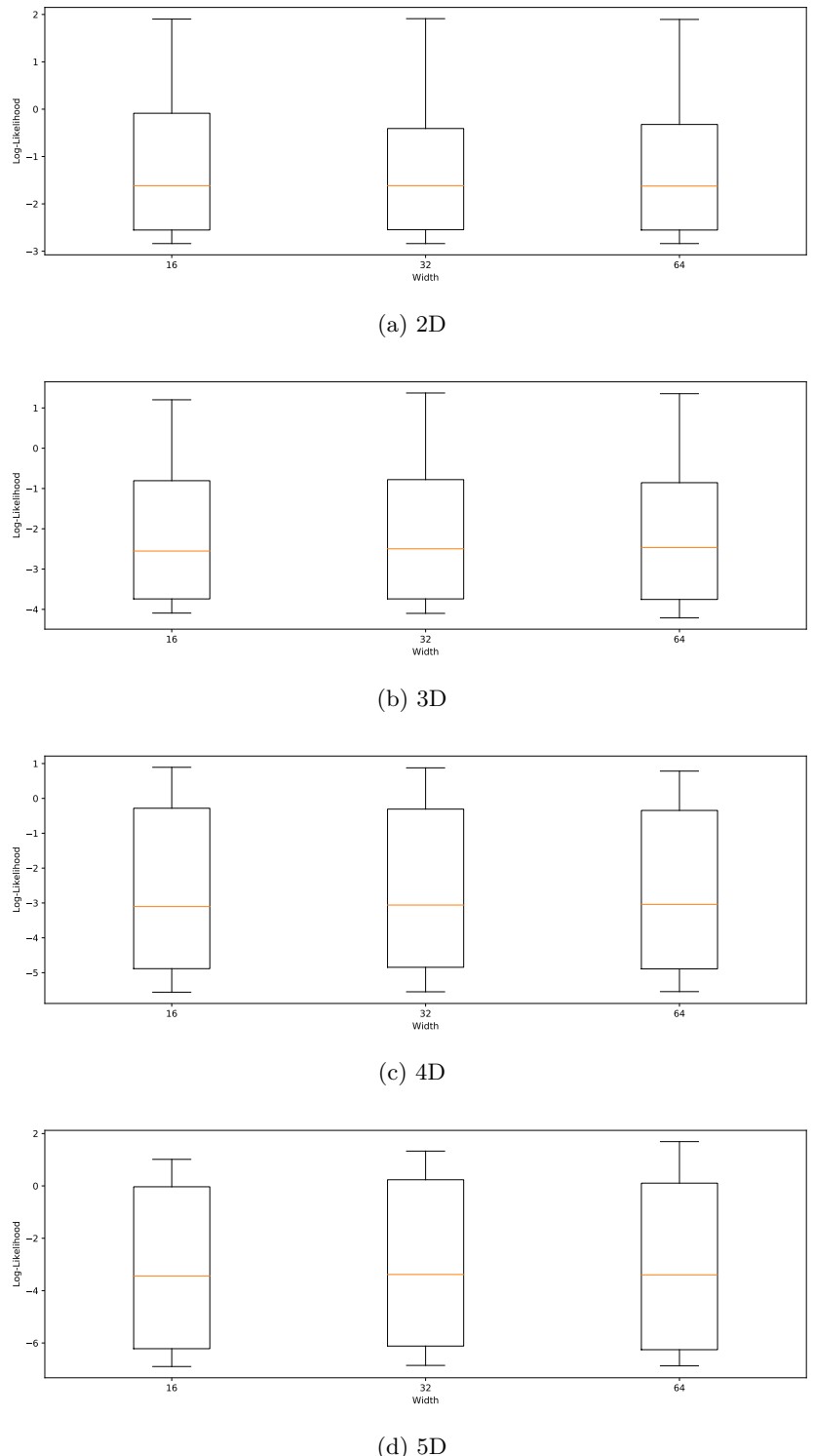

(a) 2D

(b) 3D

(c) 4D

(d) 5D

Figure 7: **Log-Likelihoods (Continuous Flows).**

# D Extended Results

In this section, we include results that evaluate our algorithm as a function of flow-architectures and their capacities. We also include plots to visually summarize our quantitative results.

| Depth | Estimator | Absolute Error | Relative Error | Width | Estimator | Absolute Error | Relative Error |
|---|---|---|---|---|---|---|---|
| 3 | IS | 0.00449±0.01048 | 0.02683±0.04063 | 16 | IS | 0.00552±0.01409 | 0.03095±0.04505 |
|  | MC | 0.00359±0.00334 | 0.04844±0.05345 |  | MC | 0.00368±0.00335 | 0.04851±0.05277 |
|  | BF-A (**ours**) | **0.00115±0.00479** | **0.01719±0.07362** |  | BF-A (**ours**) | **0.00195±0.00767** | **0.02290±0.07088** |
| 5 | IS | 0.00562±0.01381 | 0.03167±0.04787 | 32 | IS | 0.00544±0.01357 | 0.02940±0.04310 |
|  | MC | 0.00372±0.00336 | 0.04833±0.05216 |  | MC | 0.00365±0.00334 | 0.04782±0.05253 |
|  | BF-A (**ours**) | **0.00142±0.00536** | **0.02221±0.07773** |  | BF-A (**ours**) | **0.00130±0.00609** | **0.01944±0.07800** |
| 7 | IS | 0.00585±0.01499 | 0.03113±0.04583 | 64 | IS | 0.00501±0.01200 | 0.02932±0.04666 |
|  | MC | 0.00375±0.00342 | 0.04776±0.05186 |  | MC | 0.00374±0.00342 | 0.04820±0.05217 |
|  | BF-A (**ours**) | **0.00192±0.00775** | **0.02470±0.08521** |  | BF-A (**ours**) | **0.00125±0.00396** | **0.02184±0.08754** |

Table 3: **Quantitative Evaluation (Capacity - Glow and MAF).** We show the absolute and relative errors of each of the methods with a sample budget of 4,000 points as a function of the depth and width of discrete flows as explained; we see that BF-A outperforms both IS and MC.

| Flow | Method | Absolute Error | Relative Error | Width | Estimator | Absolute Error | Relative Error |
|---|---|---|---|---|---|---|---|
| Glow | IS | 0.00395±0.01131 | 0.02524±0.03818 | 16 | IS | 0.00631±0.02026 | 0.02975±0.04846 |
|  | MC | 0.00357±0.00328 | 0.05072±0.05421 |  | MC | 0.00383±0.00472 | 0.04961±0.04996 |
|  | BF-A (**ours**) | **0.00155±0.00726** | **0.01963±0.07250** |  | BF-A (**ours**) | **0.00171±0.00504** | **0.01430±0.03244** |
| MAF | IS | 0.00667±0.01480 | 0.03445±0.05032 | 32 | IS | 0.00614±0.01808 | 0.03005±0.04966 |
|  | MC | 0.00379±0.00346 | 0.04568±0.05062 |  | MC | 0.00400±0.00420 | 0.04991±0.05094 |
|  | BF-A (**ours**) | **0.00145±0.00473** | **0.02311±0.08501** |  | BF-A (**ours**) | **0.00131±0.00343** | **0.01034±0.02098** |
| FFJORD | IS | 0.00649±0.02000 | 0.03021±0.04890 | 64 | IS | 0.00701±0.02152 | 0.03080±0.04856 |
|  | MC | 0.00405±0.00455 | 0.04955±0.05034 |  | MC | 0.00432±0.00470 | 0.04914±0.05012 |
|  | BF-A (**ours**) | **0.00157±0.00417** | **0.01196±0.02529** |  | BF-A (**ours**) | **0.00170±0.00391** | **0.01132±0.02083** |
| (a) **Flow-Architecture** | | | | (b) **Capacity - FFJORD** | | | |

Table 4: **Quantitative Evaluation**. **(a)** shows the absolute and relative errors of each of the methods for a sample budget of 4,000 points as a function of the flow-architecture; we see that BF-A outperforms both IS and MC. Most interestingly, we find that our method is more efficient with Neural ODEs than with Glow or MAF – our analyses show that this can be attributed to the fact that Neural ODEs have much smoother diffeomorphic transformations than Glow and MAF. Perhaps more importantly, the Neural ODEs also obtain similar or better training log-likelihoods than Glow and MAF (Appendix C). **(b)** contains the absolute and relative errors of each of the methods with a sample budget of 4,000 points as a function of the number of hidden units of two-layer MLPs parameterizing FFJORD; we see that BF-A outperforms both IS and MC.

| Dim | Method | Time (sec) |
|---|---|---|
| 2 | IS | 0.00616±0.01209 |
|  | MC | 0.00614±0.01178 |
|  | BF-A (**ours**) | 0.00896±0.01375 |
| 3 | IS | 0.00931±0.01858 |
|  | MC | 0.00938±0.01773 |
|  | BF-A (**ours**) | 0.01380±0.02357 |
| 4 | IS | 0.01206±0.02466 |
|  | MC | 0.01193±0.02288 |
|  | BF-A (**ours**) | 0.01784±0.03092 |
| 5 | IS | 0.01708±0.03419 |
|  | MC | 0.01730±0.02887 |
|  | BF-A (**ours**) | 0.02593±0.04062 |

Table 5: **Runtime Evaluation.** We show the per-sample running times of all the estimators, averaged across all configurations, separately for each dimensionality. In order to obtain a fair evaluation, we ran all of the timing experiments in a single non-preemptible job having access to 8 CPUs, 64GB RAM and one Tesla T4 GPU (16GB).

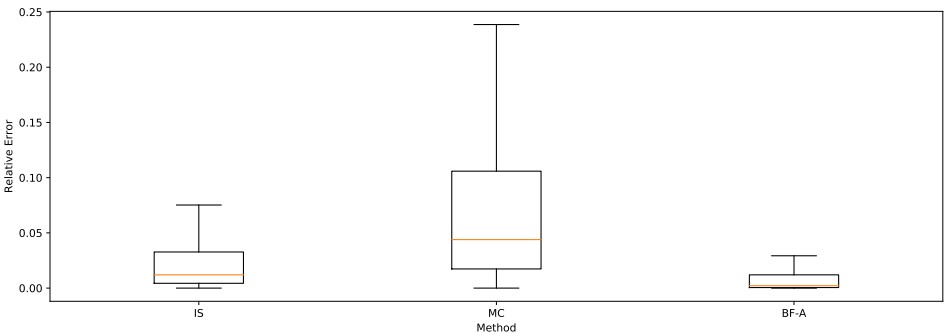

(a) Aggregate Evaluation of Estimators

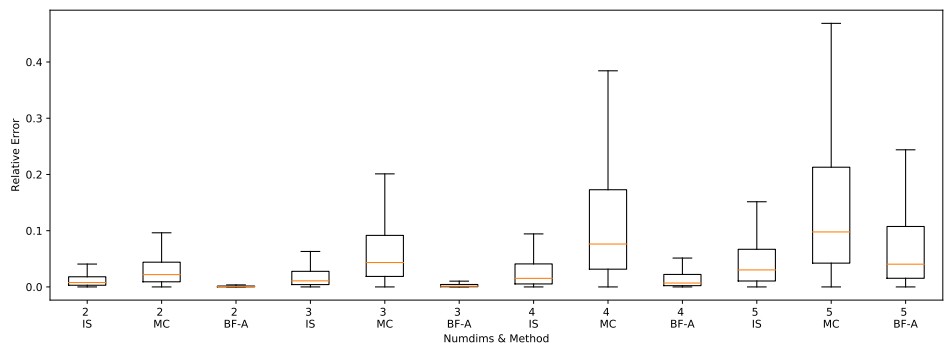

(b) Evaluation of Estimators with Growing Dimensions

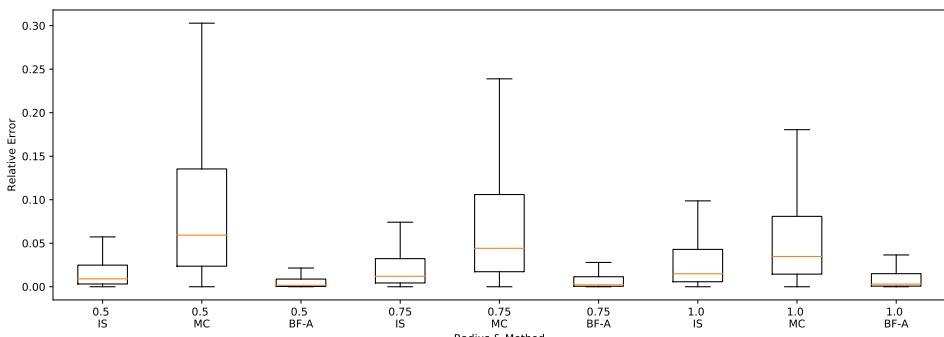

(c) Evaluation of Estimators with Growing Hull Sizes.

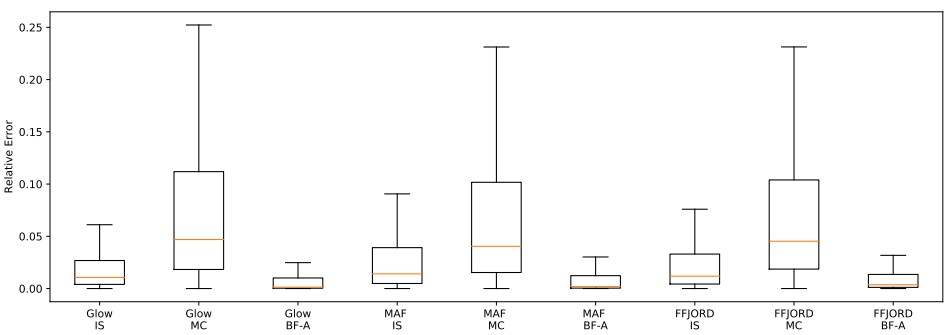

(d) Evaluation of Estimators as a function of Flow Architectures.

Figure 8: **Visualizations of Quantitative Results.**

