# OpenReview forum: "Efficient CDF Approximations for Normalizing Flows"
_TMLR — Accepted by TMLR_

### Review · Reviewer_Vy3B · 2022-06-25

**Summary Of Contributions:**

This manuscript proposes a strategy for computing cumulative distribution functions (CDFs) of complex and potentially high-dimensional probability distributions over complicated regions. The main technical innovation is found in section 4, the development of an estimator that uses samples from the base measure of a normalizing flow and propagates them through the boundary of the region of interest. The authors test their approach numerically and compare to Monte Carlo estimates and importance sampled estimates.

**Broader Impact Concerns:**

None.

**Requested Changes:**

I would say that the most pressing change is that the authors must provide a clearer articulation of how one would construct the representation of the polytope of interest. For a domain with nontrivial geometry, building the set of simplices $S_i$ seems quite complicated.

Another point the authors must address is the role of training and inaccuracy in the representation of the target distribution due to approximation by a normalizing flow.

I think it is imperative that the authors consider a domain that is not simply an arbitrarily chosen convex region around a point, but rather one that is motivated by some problem of interest.

It would be nice to see experiments on the BF-S algorithm.



**Strengths And Weaknesses:**

The paper nicely reframes the statistical estimation problem in terms of the divergence theorem and replaces an integration over a d-dimensional volume with a (d-1)-dimensional integration problem. This could lead to big improvements in performance of the estimator in low dimensional settings, but I am skeptical that it will help much in higher dimensions.

The exposition throughout is clear, and the main ideas are explained well. That said, the details of the problem could be stated more clearly.

Weaknesses:

Throughout, the paper supposes that a representation of the probability distribution of interest is given via a normalizing flow. In practice, a complex distribution coming from data will never be exactly represented in this way; training will lead to errors on the distribution that will certainly propagate to errors on the CDF.

The authors propose a stochastic algorithm, but seem only to use the deterministic one. Why is BF-S never evaluated experimentally?

The experiments go only to $d=5$.

---

> ### Author Response · Authors · 2022-07-09
> **Reply to Reviewer Vy3B**
>
> Thank you very much for your detailed review.
>
> **Domain-specific Convex Regions.** We follow previous works such as Cunningham et al. [1] in evaluating our CDF estimation algorithm on randomly drawn convex regions. In practice, convex regions can be easily obtained from an “intersection of half-spaces” definition, wherein a domain-expert can define the convex-region by specifying linear inequalities on the data dimensions. Noting that a domain-based evaluation with a specific application would likely be biased in terms of the sizes and shapes of the region, our evaluation approach was developed with an aim to account for the variance in the sizes and shapes of regions as a function of the domain and application.
>
> [1] John P. Cunningham, Philipp Hennig, and Simon Lacoste-Julien. Gaussian probabilities and expectation propagation. Journal of Machine Learning Research, 2013.
>
> **Normalizing Flows and Expressivity.** Normalizing flows are a very expressive and powerful class of generative models and many architectures are known to be universal approximators [1]. For a more detailed discussion on the expressivity of normalizing flows, please see Section 2.2 of Papamakarios et al. [2]. In our experiments, we follow the settings described in Grathwohl et al. [3] in constructing our normalizing flows with sufficient capacity to fit the spectrum of information contained in the UCI datasets. Furthermore, we also include experiments that demonstrate the efficacy of our method as a function of capacity of normalizing flows — by varying the number of hidden-units and depths.
>
> [1]  Holden Lee, Chirag Pabbaraju, Anish Prasad Sevekari, and Andrej Risteski. Universal Approximation Using Well-Conditioned Normalizing Flows. NeurIPS 2021.\
> [2] Papamakarios, G., Nalisnick, E., Rezende, D. J., Mohamed, S., and Lakshminarayanan, B. Normalizing flows for probabilistic modeling and inference. Journal of Machine Learning Research, 2021.\
> [3] Will Grathwohl, Ricky T. Q. Chen, Jesse Bettencourt, Ilya Sutskever, and David Duvenaud. FFJORD: free-form continuous dynamics for scalable reversible generative models. 7th International Conference on Learning Representations, ICLR, 2019.
>
> **Higher Dimensions.** The CDF over a compact region is defined by a computationally challenging integration over a D-dimensional volume, which inherently scales poorly with dimension. Here we describe how to exploit the diffeomorphic structure of distributions defined by normalizing flows to produce a more efficient CDF estimation method and validate our approach on problems between 2D and 5D. While higher-dimensional CDFs are certainly of interest, their computation can be challenging even in lower dimensions. As noted in our evaluation, MC algorithms would require 50k samples to match the relative errors obtained by BF-A with just 4k samples in 2D. The proposed estimator can in principle be applied to higher-dimensional problems as well, although the structural advantages compared to traditional Monte-Carlo estimators (MC, IS) might not be as pronounced. However, there is room to improve the proposed methods by exploiting more information about the transformations (e.g., with better adaptation schemes, local derivatives, etc.) and we believe this is a promising direction for future work.
>
> **BF-S Estimator.** Direct application of the BF-S algorithm is similar to the Monte-Carlo methods (MC, IS) and does not allow efficient reuse of points.  As noted in the evaluation section, the BF-S estimator has a higher variance than the related Monte-Carlo methods and accrues relative errors of about 0.19 on average with 4k samples, higher than other baselines – we will include these results in Table 1.  Note that the unbiased stochastic BF-S estimator is introduced as an intermediate step in deriving the biased adaptive BF-A estimator (compare Eq.(12) and Eq.(14)), illustrating the concept of G-field integration over the boundary. Critically, the BF-S estimator makes explicit use of properties of the input region, such as its boundary, allowing us to derive the more efficient and adaptive BF-A estimator. In future work, one could also consider stochastic-adaptive variants of the BF-S estimator and the intermediate results themselves may be useful for certain applications.

---

### Review · Reviewer_Suvx · 2022-07-03

**Summary Of Contributions:**

The paper considers the problem of building efficient estimators for CDFs using push-forward transformations, specifically normalizing flows. Towards this end, they formulate the problem of estimating a CDF over a region $\boldsymbol{V}$ in a target space as estimating the corresponding volume $\boldsymbol{W}$ in the base space. Subsequently, they take this idea to build adaptive , stochastic and unbiased estimators defining the boundary of the region in the base space more faithfully by iteratively adding points.

**Requested Changes:**

- *Writing* : I believe the paper will benefit from a notations and setup section wherein all the mathematical notations are established and the technical terms used are properly defined beforehand. Furthermore, I think the paper will also benefit from having more layman explanations accompanying useful technical derivations.

- *Empirical Analysis*: I think the empirical analysis needs to strengthen more to be able to demonstrate the utility of the proposed estimator in higher dimensions which is where I'd say the problem is harder for classical alternatives.

**Strengths And Weaknesses:**

**Strengths**:
- The problem of using diffeomorphic push-forward transformations to estimate quantities like the CDF is a nice step in exploring the utility of such probabilistic models considering their already demonstrated utility in representing conditional/ marginal distributions as well as MCMC augmented sampling from multi-modal distributions.
- The authors also did a terrific job in establishing how the CDF can be computed by relating it to the volume of the region in the base space and subsequently building the arguments to finally arrive at the chain of equivalences as shown in Fig.3 (which is very good to communicate the whole chain of arguments).

**Weakness**:
The major weaknesses of the paper are its writing (which lacks clear introduction of mathematical concepts before delving into the details) and empirical analysis to evaluate the estimator as well as comparison to baselines. I'll detail these below:

- *Empirical Analysis* :
1) Similar to Monte Carlo methods or IS, I feel the performance of the proposed estimator is dependent on how well the simplex is constructed, specifically the samples $\boldsymbol{x} \sim \mathcal{U}_{S_i}$. I wonder how the scaling here works in higher dimension since the experiments presented are on very low dimensions.
2) I also find the Monte Carlo baseline to be a weak baseline considering that in this regime a Quasi Monte Carlo baseline will be more appropriate and stronger.
3) Also, for comparison with IS and MC estimators, the authors fix a budget of samples used. However, I believe a fair comparison might be of fixing computational time and then evaluating the methods' performance when run for this fixed amount of time. Given the computational cost of constructing & refining the simplices iteratively, I am curious of its effect on run-times.
4) I also find the experiments presented to be restricted to fairly low dimensions (2-5); however, the main challenge in this space will arise in really high dimensions. I wonder how well the method would scale to high dimensions compared to classical alternatives.

- *Writing*: While, overall, the paper is well written, I believe the exposition needs to be made more accessible. For example, at the beginning of section 4, I really appreciated the authors effort to explain their potential solution in simple words (I wish more ML papers would follow this). However, I felt the effort gor diluted by the fact that the authors used quite a bit of technical jargon (that is also non-standard in ML) that was not introduced or defined before which made it hard to follow e.g. *flux* is not defined. This persists in many areas of the paper where the appropriate terms have not be introduced or defined (even though they might be standard terms in say a Physics based paper) e.g. given the plethora of mathematical terms used, it is not easily clear how Eq. 4 and Eq.5 are related. This is made more difficult by the fact that accompanying terms have not been properly defined leaving it for the reader to figure out the intuition as well as the definitions. As another example, In Lemma 4.2, how should one interpret G? The original problem was stated in terms of V. How should one connect that to the formulation here intuitively?

---

> ### Author Response · Authors · 2022-07-09
> **Reply to Reviewer Suvx**
>
> Thank you very much for your detailed review; we will upload a revision adding an intuitive definition of flux and add explanations accompanying the derivations wherever relevant.
>
> **Samples from Simplex.** The adaptive BF-A estimator acquires samples on the boundary simplices by following the splitting strategy described in Section 4.3.2 and not by randomly sampling from the simplex (${\bf x}\sim \mathcal{U}_{{\bf S}_i}$). We note that the use of this splitting strategy significantly improves both the accuracy and sample-efficiency of the CDF prediction compared to Eq.(12); future research on splitting strategies may be able to achieve even better sample efficiencies.
>
> **Quasi Monte-Carlo Estimators.** Quasi Monte-Carlo estimators are deterministic estimators that aim to position points more uniformly and regularly within a given polytope. While they are well-studied for hypercubic regions, their extension to simplicial polytopes is non-trivial (cf. [1] on applying Quasi Monte-Carlo techniques to regular shapes other than hypercubes for recent efforts in this direction). We ran experiments by choosing uniform grid sampling as Quasi Monte-Carlo method — limiting ourselves to hypercubic regions — and obtained the following results: in 2D, uniform grid sampling performs similarly to the IS estimator but has an ~13x higher relative error compared to BF-A for a sample budget of 4k points. In 3D-5D, we observed a significant decrease in performance and up to ~30x larger errors than IS on average. This is not surprising as Quasi Monte-Carlo estimators can perform arbitrarily badly for distributions whose peaks are sharp compared to the size of the region. Thus, we find that Quasi Monte-Carlo methods are not a practical or competitive CDF estimation technique and do not include them in the baselines.
>
> [1] Basu, Kinjal. Quasi-Monte Carlo Methods in Non-Cubical Spaces. Stanford University, 2016.
>
> **Higher Dimensions.** The CDF over a compact region is defined by a computationally challenging integration over a D-dimensional volume, which inherently scales poorly with dimension. Here we describe how to exploit the diffeomorphic structure of distributions defined by normalizing flows to produce a more efficient CDF estimation method and validate our approach on problems between 2D and 5D. We agree that higher-dimensional CDFs are a very challenging problem, however, their computation can be challenging even in lower dimensions. As noted in our evaluation, MC algorithms would require 50k samples to match the relative errors obtained by BF-A with just 4k samples in 2D. The proposed estimator can in principle be applied to higher-dimensional problems as well, although the structural advantages compared to traditional Monte-Carlo estimators (MC, IS) might not be as pronounced. However, there is room to improve the proposed methods by exploiting more information about the transformations (e.g., with better adaptation schemes, local derivatives, etc.) and we believe this is a promising direction for future work.
>
> **Absolute Runtime.** As the accuracy of the CDF estimates of the baseline Monte-Carlo methods (MC, IS) is mainly a function of the sample size [Cunningham et al, 2013], we made a conscious decision to compare all methods based on sample budgets. Absolute runtimes depend on a number of confounding factors – programming language, dedicated data structures, compute device, and low-level optimizations – making a fair comparison difficult. While the absolute runtime of our unoptimized Python implementation for an identical number of samples is approximately comparable to the baseline estimators, our method’s higher sample efficiency allows us to use less samples to achieve the same error.

---

### Review · Reviewer_RYPp · 2022-07-03

**Summary Of Contributions:**

**Note.** This review is being provided by the Action Editor.

This paper tackles the problem of estimating the CDF of a density described through a normalizing flow. To do so, the paper uses the divergence theorem to turn the integral over a volume into a surface integral. This theorem can be applied on either the untrasformed domain or the transformed domain; the paper argues why it is preferable to focus on the latter. The paper goes on and proposes two methods to approximate the surface integral on the transformed domain: BF-S (a stochastic unbiased approximation) and BF-A (a deterministic but biased approximation). The paper compares BF-A empirically on a number of flows and dataset against two methods: direct Monte Carlo estimator of the CDF and importance sampling, reporting smaller errors for BF-A.


**Broader Impact Concerns:**

No concerns.


**Requested Changes:**

Overall, the paper is already in good shape, although there is some room for improvement. I would appreciate if the authors could answer and/or discuss on the weaknesses listed above.

Besides that, it can be useful to add some discussion on the following points:

- Although the estimator presents lower error than MC and IS, these results are aggregated over a bunch of scenarios. It can be useful to know in which settings the estimator exhibits worse performance than the baselines, and in which settings the estimator is particularly effective.

- How sensitive are the results to the particular way in which the regions are chosen in the experiments? (i.e., starting from a sample from the model and growing a ball of a fixed radius around it, then form a hull using all samples within that ball). For example, would it be different if we started from a data sample instead?

**Minor comments.**

- I initially got a bit confused by the expression in Eqs. (8) and (9) (and Appendix A overall), as it looks like a determinant over a non-square matrix. It would help to add a comment about the size of each component in the matrix being formed.

- The notation on Section 4 and 5 becomes a bit cumbersome at times, although I don't have good suggestions on how to improve it. As a couple of minor comments, I would suggest introducing the definition of J earlier on (e.g., on Section 3), as well as using only one symbol to denote the divergence (currently the paper uses either $\nabla \cdot F$ or $\text{div}(F)$ interchangeably). That said, the Figures help a lot to keep in mind the notation that has been adopted.

- There are a few typos that need to be fixed. For example, there are a few \citet that should be \citep and vice-versa: page 2 (Genz & Bretz), page 3 (Cunningham et al.), page 8 (the tree references in the caption of Fig. 5), page 9 (Dua & Graff).

- There is a missing period (.) after "Copulas" in the paragrah in page 3.

- In Fig. 4, it would help to clarify that the two simplices are the triangles ABC and ABD.


**Strengths And Weaknesses:**

**Strenghts.**

- The paper is well motivated and the problem being addressed is relevant.

- The application of the divergence theorem for normalizing flows is not straightforward, yet the paper finds a way to form a practical estimator using it.

- The ideas are nicely presented following a natural ordering.

- The experimental analysis includes a large number of settings.

- The paper promises to release code.


**Weaknesses.**

- The estimator BF-S is not included in the experimental analysis of Section 5, leaving the open question of how it performs in practice. If it does not perform well in practice, consider removing it from the paper (or relegating to an Appendix). If it does perform well, it'd be interesting (and useful to the community) to know about it.

- The comparison to previous estimators is based on keeping the sample budget fixed. However, my understanding is that MC and IS are both faster than BF-A given the same sample budge, since BF-A requires additional computations and an iterative refinement procedure. Therefore, I think it would be fairer to compare estimators under the same computation budget, as opposed to sample budget.

- It is unclear why convex hulls with CDF smaller than 0.01 are rejected in the experiments. Could the authors clarify this?

- The experiments consider a low-dimensional setting ($k=2, 5$) (minor side note: why is this a change of notation from $d$ to $k$?). My understanding is that any CDF estimator operating in higher dimensions would inevitably fail due to floating point precision, raising the question of how to obtain a reliable estimator of log CDF. At the very least, a discussion about this point should be included under "Limitations", ideally mentioning the maximum dimensionality that the estimator can handle.

---

> ### Author Response · Authors · 2022-07-09
> **Reply to Reviewer RYPp**
>
> Thank you very much for your detailed review.  We are currently working on a revision which incorporates parts of this response.
>
> **BF-S Estimator.** Direct application of the BF-S algorithm is similar to the Monte-Carlo methods (MC, IS) and does not allow efficient reuse of points.  As noted in the evaluation section, the BF-S estimator has a higher variance than the related Monte-Carlo methods and accrues relative errors of about 0.19 on average with 4k samples, higher than other baselines – we will include these results in Table 1. Note that the unbiased stochastic BF-S estimator is introduced as an intermediate step in deriving the biased adaptive BF-A estimator (compare Eq.(12) and Eq.(14)), illustrating the concept of G-field integration over the boundary. Critically, the BF-S estimator makes explicit use of properties of the input region, such as its boundary, allowing us to derive the more efficient and adaptive BF-A estimator. In future work, one could also consider stochastic-adaptive variants of the BF-S estimator and the intermediate results themselves may be useful for certain applications.
>
> **Absolute Runtime.** As the accuracy of the CDF estimates of the baseline Monte-Carlo methods (MC, IS) is mainly a function of the sample size [Cunningham et al, 2013], we made a conscious decision to compare all methods based on sample budgets. Absolute runtimes depend on a number of confounding factors – programming language, dedicated data structures, compute device, and low-level optimizations – making a fair comparison difficult. While the absolute runtime of our unoptimized Python implementation for an identical number of samples is approximately comparable to the baseline estimators, our method’s higher sample efficiency allows us to use less samples to achieve the same error (e.g., in 2D, the Monte-Carlo estimator would require 50k samples to match the relative errors obtained by BF-A with just 4k samples).
>
> **Rejecting Convex Hulls.** We observed that the inclusion of very small convex hulls can lead to inconsistent results across runs due to large relative errors of the sampling-based estimator baselines (MC, IS). For fairness and to increase stability and reproducibility we reject convex hulls with CDFs smaller than 0.01; we will clarify this in the paper.
>
> **Higher Dimensions.** The CDF over a compact region is defined by a computationally challenging integration over a D-dimensional volume, which inherently scales poorly with dimension. Here we describe how to exploit the diffeomorphic structure of distributions defined by normalizing flows to produce a more efficient CDF estimation method and validate our approach on problems between 2D and 5D. The proposed estimator can in principle be applied to higher-dimensional problems as well, although the structural advantages compared to traditional Monte-Carlo estimators (MC, IS) might not be as pronounced. However, there is room to improve the proposed methods by exploiting more information about the transformations (e.g., with better adaptation schemes, local derivatives, etc.) and we believe this is a promising direction for future work.
>
>
> **Limitations and Fine-grained Experiments.** In order to study how our method performs with increasing complexity, we experiment with growing dimensionality, hull sizes, and model capacity. We note that the main limiting factor is the dimensionality as we have discussed above. Appendix D contains more fine-grained results (e.g. model capacity and model architecture) and visualisations of all experiments. We will upload a revision adding a discussion on scalability to the limitations section.
>
> **Sensitivity.** After excluding small regions (as discussed above) we find that the results are reproducible across runs and our experiments explore several different ways of generating convex hulls with the intention of exploring different limits of the method behaviour. Computing convex hulls around data points is similar to computing convex hulls around samples from the model in that data points and samples both generally lie in the typical sets of the distribution assuming the normalizing flow was well fit to the data distribution.

---

> > ### Comment · Reviewer_RYPp · 2022-07-26
> > **Thank you. Some further discussion points**
> >
> > Dear authors,
> >
> > Thank you very much for your detailed response. I appreciate the paper has been noticeably improved.
> >
> > Upon reading your responses and the updated version of the paper, I would like to suggest adding some more discussion:
> >
> > + Re absolute runtime: I agree that there are a number of confounding factors, yet this is true for virtually any algorithm. Given that your current (unoptimized) implementation runs comparably as fast as the baselines with the same number of samples, this comparision can be included in the paper, explicitly acknowledging that the exact running times may vary depending on the implementation and hardware. At the very least, some discussion on running time should be included in the text.
> >
> > + Please include a brief discussion about the approximation errors in Eqs. (13)-(14) due to the vector field being non-constant along the faces of the polytope.
> >
> > + In Tables 1 and 2, the error intervals intersect. Would it be possible to run some statistical tests to assess the significance of the results?
> >
> > + Minor issue: Please replace the notation $d^{-1} x$ with $x / d$ as the former is confusing.

---

### Author Response · Authors · 2022-07-13
**New Revision**

We thank all reviewers for their thoughtful and detailed feedback. We have uploaded a new revision of our paper incorporating the following changes (highlighted in blue):
- [Suvx] Added an intuitive definition of flux at the beginning of section 4.
- [Suvx] Expanded Eq. (5) to show how the divergence theorem is applied.
- [Suvx] Added a note referencing Fig. 1 (before subsection 4.2.1) explaining how one should interpret $\bf G$.
- [Suvx and RYPp] Introduced the Jacobian $J$ in Section 3. With this new addition, section 3 now introduces all the basic notations and definitions: the volumes $\bf V$ and $\bf W$, the flow transformations $f$ and $g$, and the Jacobian $J$.
- [RYPp] The paper now uses $\nabla \cdot (\cdot)$-notation everywhere to denote divergence.
- [RYPp] Added notes about the shapes of the vectors/matrices before Eq.(8) and revised the determinant notation, both in the main text and in the Appendix.
- [RYPp] Clarified the simplices in Fig. 4, explained the rationale behind rejecting convex hulls with CDFs smaller than 0.01, and corrected the typos related to \citep and \cite.
- [Vy3B, RYPp] Added the BF-S result to table 1.
- [Vy3B, Suvx, RYPp] Expanded the limitations section to clarify the advantages of our estimator in low-dimensional problems and opportunities to extend the scalability of our estimator in future work.

Please let us know if there are any remaining concerns.

---

### Decision · Action_Editors · 2022-07-26

**Recommendation:** Accept with minor revision

**Comment:**

Thank you for your submission. The paper proposes a novel way to estimate the CDF of a distribution modeled by a normalizing flow. The main technical contribution is a clever application of the divergence theorem. Although the paper has some limitations, the paper is technically correct and all three reviewers lean towards acceptance. Moreover, the main limitation (lack of scalability with dimensionality) has been properly discussed in the paper. Thus, my final recommendation is "Accept with minor revision".

As for the final version, I expect the authors to include some discussion about the absolute runtime, the approximation errors, and the error intervals in Tables 1 and 2.

---

> ### Author Response · Authors · 2022-08-21
> **Camera-ready Revision**
>
> We want to thank all reviewers again for the valuable comments and suggestions that have contributed to this camera-ready version of our paper. This update addresses the remaining requests:
> * We added a discussion on absolute runtimes in Section 5 (Discussion) and summarized our results in Table 5.
> * We ran statistical tests to confirm that the improvements brought about by our method are statistically significant and updated the captions of Table 1 and Table 2 accordingly.
> * We updated Section 4.4.1 to explain the approximation errors.
>
> We thank everyone for a swift and smooth review process.